# Collaborative Response of the Host and Symbiotic Lignocellulytic System to Non-Lethal Toxic Stress in *Coptotermes formosanus* Skiraki

**DOI:** 10.3390/insects12060510

**Published:** 2021-05-31

**Authors:** Wenhui Zeng, Bingrong Liu, Wenjing Wu, Shijun Zhang, Yong Chen, Zhiqiang Li

**Affiliations:** Guangdong Key Laboratory of Animal Conservation and Resource Utilization, Guangdong Public Laboratory of Wild Animal Conservation and Utilization, Institute of Zoology, Guangdong Academy of Sciences., No. 105, Xingang Xi Road, Guangzhou 510260, China; zengwh@giabr.gd.cn (W.Z.); liubr@giabr.gd.cn (B.L.); Wuwj@giabr.gd.cn (W.W.); Zhangsj@giabr.gd.cn (S.Z.)

**Keywords:** lignocellulolytic system, non-lethal stress, symbiont, termite, toxic tolerance

## Abstract

**Simple Summary:**

*Coptotermes formosanus* Shiraki is a wood feeding lower termite and is widely distributed in many areas. The dynamic adjustment of the *C. formosanus* digestive system to unfavorable survival environments was investigated via non-lethal toxic feeding. The toxic stress did not change the dominant role of microbial lignocellulases in cellulose degradation of *C. formosanus*. The core symbiotic community was stable in abundance during the tolerance to the toxic treatment. However, a large number of low abundance taxa were significantly enriched by the low toxic feeding. These rare bacterial lineages likely contribute to toxic stress tolerance of termite. Above all, these findings add important new knowledge to our understanding of environmental adaptation of the lignocellulose hydrolysis system in termites.

**Abstract:**

Disturbing the lignocellulose digestive system of termites is considered to be a promising approach for termite control. The research on the tolerance mechanism of the termite lignocellulose digestive system to harmful environment conditions is limited. In this study, we keep *Coptotermes formosanus* Skiraki under a non-lethal toxic condition by feeding the termites with filter paper containing the kojic acid (a low toxic insecticide). The effects of low toxic stress on the activities and gene expressions of host/symbiotic originated lignocellulases, and on the symbiotic microbial community structure of *C. formosanus* were explored. Our result showed that the low toxic stress would lead to the synchronous decrease of cellulase and hemicellulase activities, and supplementary increase of corresponding gene expressions. The symbiotic community maintained its role as the main force in the lignocellulolytic system of *C. formosanus*. Meanwhile, a large number of rare taxa were significantly enriched by kojic acid treatment. These numerically inconspicuous bacterial populations might be responsible for the functions similar to phenoloxidase or insecticide detoxification and enable *C. formosanus* to tolerate the harmful environment. Overall, our data suggested that the digestive adaptation of *C. formosanus* to physiotoxic feeding is closely related to the triple collaboration of termites–flagellates–bacteria.

## 1. Introduction

Termites are one of the most important economic pests in the world due to their efficient degradation of lignocellulose [1,2]. Elucidating the lignocellulolytic related mechanism in termites is very important for the development of cellulase-specific biopesticides and biomass conversion [3].

The lower termite possesses a synergistic dual degradation system that digests these macromolecular substances consisting of symbiotic fauna (protists and bacteria), which provide exogenous activities in the hindguts of workers, and the endogenous activity from the foregut/salivary gland and midgut of the host [4,5]. *Coptotermes formosanus* Shiraki is a wood feeding lower termite and is spread widely throughout southern China [6]. *C. formosanus* harbours four species of flagellates, which have been identified as *Pseudotrichonympha grassii* Koidzumi, *Holomastigotoides hartmanni* Koidzumi and *Holomastigotoides minor* Nishimura and *Cononympha leidyi* Koidzumi [7]. Dominant cellulases (glycosyl hydrolase family, GHF5, 7) and hemicellulases (GHF10, 11) in the hindgut of *C. formosansus* are encoded by flagellates *P. grassii* and *H. mirabile* [8], which play a dominant role in cellulose/hemicellulose decomposition [9,10,11]. In *C. formosanus*, more than 70% of prokaryotic microbes live within the protist cell and the rest either attach to the protist surface or occur freely in the alimentary fluid of the termite [12]. The major functions of lower termite symbiotic bacteria are the performance of multiple metabolic processes, such as nitrogen cycling/nutrients synthesis, carbohydrate metabolism and energy metabolism [13,14]. In recent years, some groups of bacteria were found to participate in the metabolism of physiologically toxic substances, such as intermediates of lignin and other low molecular weight phenols [15,16,17,18,19,20]. The endogenous digestive genes of *C. formosanus* encode cellulases, predominantly GHF1 and GHF 9 [8], which digest a part of the cellulosic materials on the surface of the wood particles [10,21]. In addition, most of the auxiliary redox enzymes (such as peroxidases, esterases and laccases) thought to be involved in lignin decomposition, are endogenously secreted [19,22]. Therefore, the survival of lower termites feeding on lignocellulose depends on the effective cooperation of host termites and hindgut symbiotic digestive systems. With this respect, disturbing or disrupting the synergistic dual-digestive system is considered a promising approach for lower termite control. Despite termiticide being widely used, a few studies concerning the cellulase inhibitors of termite are limited to the inhibition efficiency of the target cellulase activity [23,24]. The tolerance mechanism of the lignocellulitic digestive system of termites to physiologically toxic substances still needs to be explored.

Kojic acid is a mycotoxin and an inhibitor of phenoloxidase activity (tyrosinase and laccase) with low toxicity to insects [25,26,27]. Based on the low toxicity of kojic acid to insects and an optimized feeding concentration, we put the *C. formosansus* under a non-lethal toxic stress for ten days. Under the premise of maintaining the normal activity and food intake of termites, the phenoloxidase activity of *C. formosanus* was inhibited by adding kojic acid to termite feeding filter paper. We investigated variations in the relationship between the activity and gene expression of endogenous and symbiotic ligonocellulases. In addition, the number of flagellates and the distribution and structure of the bacterial communities were analyzed to study how the gut microbial community synchronously correspond. This study will provide important insights into the collaborative tolerance mechanism of the dual-lignocellulolytic system in lower termites to harmful environmental stress.

## 2. Materials and Methods

### 2.1. Termite Collection, Kojic acid Treatment and Feeding Assays

The termite *C. formosanus* used in this study were obtained from five laboratory-maintained colonies (collected from Guangzhou City in Guangdong Province, China). Termites were reared under laboratory conditions at a temperature of 27 ± 1 °C, 75 ± 5% relative humidity in complete darkness, and were starved for 18 h before use. The starved termites were separated into two treatment groups (80 workers and two soldiers per group): one group (treatment group) was fed on filter paper soaked with 20 mM kojic acid aqueous solution (KOJ, Aladdin, Shanghai, China) and the other group (control group) was fed on filter paper soaked with sterile distilled water (dH_2_O).

The richness, diversity or structure of microbial community in the hindgut of lower termites also can rapidly respond to artificial feeding variation within 7 days [28,29]. If the *C. formosanus* cannot adapt to the food ingredient, their vitality or food intake will be significantly reduced within a week, which will cause shriveling of the abdomen and a large number of deaths. Thus, for each of the groups, the termites were placed into separate petri dishes (9 cm diameter) and maintained in incubators for 10 d at 27 ± 1 °C and 75 ± 5% relative humidity. Each group was supplemented with 200 μL of distilled water every two days. Dead termites were removed every day and the mortality rate was calculated on the last day. The filter papers from two groups were dried at 60 °C for 18 h and weighed after drying. There were five biological replicates per group for mortality and analysis of consumption rate. The experimental kojic acid concentration (20 mM) was optimized by preliminary feeding and mortality analysis. The highest concentration of kojic acid that had no significant impact on termite mortality and consumption rate after 10 day treatment was selected for following the non-lethal toxic stress study (Appendix A).

### 2.2. Crude Enzyme Extraction and Protein Concentration Assays

Treated termites from three biological replicates of each group were washed with distilled water. Extracts of crude cellulases/hemicellulases from termite intestines were prepared according to the method described by Zeng et al. [30]. The anterior part of the termite digestive system concentrates host (termite) cellulases, and the hindgut concentrates all symbiotic fauna and their enzymes. Briefly, twenty termites were collected randomly from each replicate, immobilized on ice, and their alimentary canals dissected to separate salivary gland, foregut and midgut from the hindgut. The tissues were then homogenized in ice-cold SAB buffer (0.1 M sodium acetate, pH 5.6) before being centrifuged at 12,000× *g* at 4 °C for 15 min. The resultant supernatants were transferred to fresh tubes and used as crude enzyme for measuring cellulase activity. For phenoloxidase activity measurement, the termite tissues were homogenized in ice-cold PBS buffer (0.01 M phosphate, pH 7.2) and the steps used for the preparation of crude cellulase were repeated to obtain a crude enzyme for measuring phenoloxidase activity. Protein concentrations were determined by using the Bradford Protein Assay kit (Solarbio, Beijing, China).

### 2.3. Enzyme Activity Assays

Three major cellulases: endo-β-1, 4-glucanase (EC; EG. 3.2.1.4), exo-β-1, 4-cellobiohydrolase (CBH; EC. 3.2.1.91) and β-glucosidase (BG; EC. 3.2.1.21), which cooperate to completely degrade cellulose [5,31], and one dominant hemicellulase, endo-β-1, 4-xylanase (EX, EC. 3.2.1.8), which hydrolyzes the xylan backbone [9], were selected to evaluate the hydrolytic ability of termites. EG and BG activity were assayed by measuring the release of glucose from 1% carboxymethyl cellulose sodium (CMC) and 1% salicine, for the respective enzymes, and EX activity was assayed by measuring the release of xylose from 1% xylan. The dinitrosalicylic acid (DNS) method was used to detect the enzymatic reaction. Glucose and xylose production were detected colorimetrically with a Victor 3 Multi-label Microplate Reader (Perkin Elmer, Waltham, MA, USA) at 540 nm using glucose as the standard. CBH activity was assayed by measuring the release of p-nitrophenol (pNP) from 1 mM p-nitrophenyl-β-D-cellobioside (pNPC), and the formation of pNP was detected colorimetrically at 405 nm. The reaction mixtures consisting of 12 μL of crude enzyme and 120 μL of 1% CMC, 1% salicine, 1% xylan or 1% pNPC were incubated at 37 °C for 60 min. One unit (U/mg protein) of lignocellulase activity was defined as the amount of enzyme capable of releasing 1 mg reducing sugar/μmol pNP per minute under the experimental conditions described [30,32].

For better correlatation of the inhibition effect with lignocellulolytic variations, the phenoloxidase activity was measured according to the method of *C. formosanus* laccase described by Geng et al. [22], and 20 mM pyrocatechol (0.01 M phosphate buffer, pH 7.2) was used as a substrate for activity measurement. The reaction mixture consisted of 20 μL enzyme preparation and 180 μL substrate solution. All reactions were carried out at 37 °C for 15 min and enzyme activity was measured kinetically at 405 nM on a microplate reader (Perkin Elmer, Waltham, MA, USA). One unit (U/mg protein) of phenoloxidase activity was defined as a 0.005 change in absorbance at 405 nm per minute per mmole enzyme.

### 2.4. Total RNA Isolation and Quantitative Real-Time PCR

Total RNA was extracted from 10 workers collected from three replicates of each kojic acid and dH_2_O group using a Total Isolation System kit (Promega, Beijing, China). Equal quantities of RNA (1.0 μg) were used as a template to generate cDNA using the PrimeScript™ RT reagent kit with gDNA Eraser (TaKaRa, Tianjin, China). The primers used in this study are designed by Lasergene version 7.1 (DNASTAR, Madison, WI, USA) and details are shown in Table 1. Quantitative real-time PCR (qPCR) was performed with Luminaris HiGreen Fluorescein qPCR Master Mix (Thermo Fisher, Shanghai, China) according to the manufacturer’s instructions. The amplification conditions were 30 s at 95 °C followed by 40 cycles of 5 s at 95 °C, 30 s at 55 °C and 30 s at 72 °C. The dH_2_O control group was used as calibrator in calculation of relative expression amount. The relative expression levels for specific genes, in relation to the reference gene *hsp-70* [30,33], were calculated by the 2^−ΔΔCT^ method [34]. Each assay was conducted in triplicate.

### 2.5. Microscopic Examination and Counting of Protozoa

To estimate protist abundance, nine hindguts from each of the three replicates of each treatment group were pulled out from the termites’ posterior ends and torn into pieces by forceps. The hindguts were not pooled and the contents of each hindgut were suspended in 10 μL PBS (20 mM phosphate buffer, pH 7.4) before gentle maceration to facilitate the release of the protozoa. Protist cells were then counted and identified using hemacytometer (SAIL BRAND, Yancheng, China) under light microscope (Nikon Elipse 80I, Tokyo, Japan). *Holomastigotoides hartmanni* and *H. minor* are difficult to distinguish by morphological characters under light microscope, the cells of *Holomastigotoides* were counted as *Holomastigotoides* spp.

### 2.6. High-Throughput Sequencing of 16S rRNA Gene Library and Sequence Processing

The hindguts of 25 workers from each of the three replicates of each treatment group were dissected using a sterile scalpel. The metagenomic DNA of hindgut microbes were extracted using the TaKaRa MiniBEST Universal Genomic DNA Extraction Kit Ver.5.0 (TaKaRa, Osaka Japan) according to the manufacturer’s instructions with slight modifications. The V4 hypervariable region of the bacterial 16S rRNA gene was amplified using the 515F (5′-GTGCCAGCMGCCGCGGTAA-3′) and 806R (5′-GGACTACHVGGGTWTCTAAT-3′) primers [35]. Sequencing was performed on the IonS5TMXL platform at Novogene Bioinformatics Technology Co., Ltd., Beijing, China. Quality filtering of the raw tags was performed under specific filtering conditions to obtain high-quality clean tags according to the QIIME (Version 1.7.0) quality-control process. Then, the paired-end reads were merged into single, longer sequences using FLASH Version 1.2.7 [36]. The major parameters included for merging the paired end reads are “Maximum Mismatch Rate of Overlapping Areas” (<0.1 bp) and “Minimum overlap region” (>10 bp). Quality filtering on the raw tags was performed under specific filtering conditions to obtain high-quality clean tags according to the QIIME (Version 1.7.0) quality-controlled process. The final effective data were generated after chimeric sequences were removed using the UCHIME2 algorithm to detect chimera sequences (http://www.drive5.com/usearch/manual/chimera_formation.html accessed on 10 May 2021), the chimera sequences were then removed. The original sequencing data was submitted to the NCBI database as a file under accession number PRJNA627843.

### 2.7. Bacterial Community Composition and Structural Analysis

The sequences with ≥97% similarity were assigned to the same operational taxonomic unit (OTU) using Uparse software (Version 7.0.1001). Representative sequences from each OTU were screened for further annotation. For each representative sequence, the GreenGene Database was used with the RDP classifier (Version 2.2) to annotate bacterial taxonomic information.

The taxa abundance of each sample was classified into phylum, class, order, family and genus levels. All of the analyses regarding clustering and alpha (within sample) and beta (between samples) diversity were performed with QIIME (Version 1.70) and displayed with R software (Version 2.15.3, Foundation for Statistics Computing, Vienna, Austria). The alpha diversity analysis included observed species, ACE (Abundance Coverage-based Estimator) and Chao1 estimators, Simpson and Shannon diversity indices and phylogenetic diversity (PD) whole tree’s Index and Good’s estimate of coverage. The Tukey test was performed to analyze the significance in alpha diversity indexes above between kojic acid treatment and dH_2_O control groups. The beta diversity analysis included Anosim (Analysis of similarities) and MRPP (Multiple Response Permutation Procedure) analyses. Metastats and *t*-tests were performed to identify the bacterial taxa that were significantly different between the kojic acid treatment and dH_2_O control groups at several taxonomic levels [37].

### 2.8. Statistical Methods

The differences in feeding amount, enzyme activities, qPCR data, and numbers of protozoa between the two treatment groups were analyzed using the independent samples *t*-test with SPSS 17.0 (SPSS Inc., Chicago, IL, USA). Data were presented as the mean standard error of the mean (SEM).

## 3. Results

### 3.1. Effect of Kojic Acid Treatment on the Physiological State of Termites

The feeding amount and mortality were used to evaluate the effects of kojic acid (20 mM) treatment on physiological state of termites. There were no significant differences in the feeding amount and the mortality between the two treatment groups (*p* = 0.07 and *p* = 0.75, respectively), indicating that normal feeding activity and survival was maintained in kojic acid treated termites (Table 2). The whole body and endogenous PO activities in the kojic acid treated group were significantly lower (*p* = 0.033 and *p* = 0.019, respectively) than those in the control group, which confirmed the inhibitory effect of kojic acid.

### 3.2. Variation in Cellulase Activities

The lignocellulolytic ability was assessed by measuring the enzyme activities of the endogenous and symbiotic lignocellulases in the *C. formosanus* digestive tract. Except for EG, both the endogenous and symbiotic lignocellulase activities were generally depressed by feeding kojic acid, especially the activities of the symbiotic lignocellulases. The symbiotic activities of BG (*t* = 8.26, *p* = 0.00), CBH (*t* = 5.25, *p* = 0.01) and EX (*t* = 10.14, *p* = 0.00) were significantly lower in the kojic acid treatment than in the dH_2_O control group (Figure 1A). In terms of the endogenous enzymes, only the EX activity was significantly depressed by kojic acid treatment (*t* = −4.14, *p* = 0.01) (Figure 1B). The order of the inhibitory effects of kojic acid on four types of lignocellulases was EX > BG > CBH > EG (Figure 1).

### 3.3. Variation in the Levels of Transcription of the Lignocellulolytic Genes

We correlated lignocellulase expression from the host and protozoa with corresponding lignocellulolytic activities using the relative transcript levels of six genes measured by qRT-PCR. Termites from the dH_2_O treatment were used as controls for gene expression analysis. In accordance with the PO activity results, the kojic acid significantly reduced the transcriptional expression of lac (*p* = −114.37, *p* = 0.00) by approximately 35-fold relative to the control group. In contrast to the patterns of variation in enzyme activities, the relative transcription levels of cellulase/hemicellulase genes were upregulated in the kojic acid treatment group, among which *bg* (*t* = 16.64, *p* = 0.00), *Peg* (*t* = 3.96, *p* = 0.02), *Pcbh* (*t* = 2.87, *p* = 0.04) and *Pex* (*t* = 13.32, *p* = 0.00) were significantly upregulated. The general level of upregulation of endogenous hydrolase genes was lower than that of the exogenous cellulase genes, which was consistent with the inhibitory relationship between endogenous and exogenous hydrolase activity (Figure 2).

### 3.4. Effect of Kojic Acid Treatment on Population Size of Flagellates

The effects of 10 days of kojic acid treatment on the population size of four species protists were investigated to further evaluate the correlation between the exogenous lignocellulase activity and gene expression. The kojic acid treatment enhanced the population size of *Holomastigotoides* spp. (*t* = 0.95, *p* = 0.40) and *S. leidyi* (*t* = 4.03, *p* = 0.02), but reduced the number of *P. grassii* (*t* = −1.92, *p* = 0.13) (Figure 3).

### 3.5. Effect of Kojic Acid Treatment on the Symbiotic Bacterial Community of the Hindgut

We analyzed the effects of kojic acid treatment on the bacterial community of *C. formosanus* in terms of bacterial diversity, community structure and the specific responses in bacterial pools. A total of 1,836,087 effective tags were obtained, which were clustered into OTUs at the 97% identity level. The rarefaction curves (Figure 4) were near saturation, and the good-coverage rate of each sample exceeded 99% (Table 3). This information indicated that the bacterial communities were sufficiently sampled for this study.

There was no significant difference in the number of effective sequencing tags between kojic acid treatment group and control group (Figure 5). The observed species number (*p* = 0.005), OTU number (*p* = 0.02), Chao1 (*p* = 0.008), AEC (*p* = 0.008) and PD tree indices (*p* = 0.004) of the kojic acid treatment group were all significantly higher than those of the dH_2_O control group (Table 3, Figure 5). These results suggest that the kojic acid treatment for ten days significantly increases the diversity of the bacterial community of *C. formosanus*.

In addition, the Anosim and MRPP analyses based on the Bray–Curtis distance were used to evaluate the distance of the bacterial community structure between the two treatment groups. Both the Anosim (*R* = 0.516, *p* = 0.1) and MRPP (*A* = 0.128, *p* = 0.1) tests indicated that the within-treatment difference was significantly smaller than the with-treatment difference, although kojic acid treatment did not reshape the bacterial community structure significantly.

The top five phyla in terms of abundance of the bacterial communities in the two treatment groups were common and accounted for 87.9% (control) and 72.58% (kojic acid treatment) of the total reads in each treatment group, with a similar order in terms of relative proportion (Bacteroidetes > Spirochaetes > Firmicute > Proteobacteria > Actinobacteria) (Figure 6). At the low taxonomic level, most of the reads were clustered in the genus *Candidatus Azobacteroides* (order Bacteroidales/phylum Bacteroides) and the genus *Treponema* (order Spirochaetales/phylum Spirochetes). These results further suggested that kojic acid treatment did not change the main structure of the bacterial community (Figure 7).

To further identify the bacterial taxa (at the phylum, class, order, family and genus levels) that were significantly altered between the two treatment groups, the taxa with relative abundance ≥0.1% (major taxa) were analyzed using *t*-tests (*p* < 0.05) and Metastat (*p* < 0.05). This threshold allowed us to retain the highest number of taxa for meaningful comparisons and to eliminate most of the extremely rare taxa in the analysis. In comparison with the dH_2_O control, the abundance of 67 and 3 bacterial taxa, which are distributed in 7 phyla, was significantly increased and decreased, respectively, after the kojic acid treatment. At the phylum level, the abundance of three core bacterial lineages (Spirochetes, Firmicute and Proteobacteria) and three low abundant bacterial lineages (Verrucomicrobia, Acidobacteria and Chloroflexi) were significantly changed by the kojic acid treatment (Table 4). Proteobacteria had the largest number of taxa (*n* = 27) with significantly increased relative abundances following kojic acid treatment. Furthermore, these 27 taxa were mainly clustered in class Alphaproteobacteria and class Gammaproteobacteria. The phylum Firmicute, which had the largest change in absolute abundance, was significantly more abundant (*p* = 0.045) in the kojic acid treatment group (9.27 ± 2.03%, average for 6708 tags) than in the dH_2_O control group (3.43 ± 0.32%, average for 2673 tags). In addition, the major abundance increase of this phylum was accounted for by the Clostridia class and family Enterococcaceae (order Lactobacillales), both of which were significantly enriched (*p* = 0.042, *p* = 0.031) by kojic acid treatment. Two rare bacterial phyla, *Acidobacteria* and *Chloroflexi*, were significantly enriched by the kojic acid treatment, with more than 425-fold (*p* = 0.024) and 760-fold (*p* = 0.008) increases in relative abundance compared with the dH_2_O control group. Only the two most abundant phyla, Bacteroidetes and Spirochete, were downregulated after kojic acid treatment. The genera *Candidatus Azobacterioides* (Bacteroidetes) and *Treponema* (Spirochete) accounted for the largest declines in the abundance (39.12 ± 2.11%, 21.76 ± 1.96% and 48.70 ± 8.19%, 29.10 ± 6.55% in the kojic acid treatment and dH_2_O control groups, respectively), although no significant differences were detected in either of the genera (Appendix A, Table 4 and Figure 7).

Generally, a 20 mM kojic acid treatment for 10 days did not significantly reshape the bacterial community of *C. formosanus*, although a large number of low abundance bacteria (relative abundance < 1%) was significantly enriched.

## 4. Discussion

### 4.1. Differential Response of Dual-Origin Cellulolytic Enzymes to Kojic Acid Treatment

After 10 days of kojic acid treatment, almost all the activities of the detected cellulases and hemicellulases derived from termites and hindgut microorganisms decreased. Furthermore, compared with the situation in which the activity of three exogenous hydrolases was significantly decreased, the endogenous glycosyl hydrolyzing ability was much less sensitive to kojic acid. The filter paper is pure cellulose, which can be completely degraded by the combination of EG, BG and CBH [10,19]. Among the three major cellulase types, both the EG and CBH activity were distributed mainly in the hindgut of the two treatment groups. Although *C. formosanus* itself have the capacity to achieve a certain amount of cellulose degradation, most of the cellulases are broken down by the dense assemblage of symbionts in the hindgut [10,13]. Consequently, the rebounding-increase in the magnitude of the exogenous hydrolase gene expression levels (approximately 3–14-folds compared with the control group) was also greater than that of the endogenous hydrolase (approximately 3–6-fold compared with the control). Thus, the differential variations in cellulase activities and corresponding gene expression in response to kojic acid treatment reflected that the non-lethal toxic stress did not change the dominant role of intestinal lignocellulases in cellulose digestion of *C. formosanus*.

Among all the hydrolases detected, the activity of exogenous EX (*p* = 0.001) significantly decreased most following the kojic acid treatment. In addition, although hemicellulase activity was distributed mainly in the hindgut, the endogenous hemicellulase activity was also significantly decreased by kojic acid treatment (*p* = 0.014). Lignocellulose is composed of approximately 40% cellulose, 25% hemicellulose and 20% lignin [38]. The substrate of EX is xylan, which is the backbone and major component of hemicellulose [5]. Although the secretion of various lignocellulases in termites is dependent on the dietary ingredients [1,19,39,40,41,42], filter paper has been shown to be more effective than xylan in inducing hemicellulose activity of termites [1,43]. Compared with other tested genes, *Pex* transcriptional amount showed a maximum supplementary increase (*p* = 0.000) after its activity was greatly inhibited, which also indicated that the hemicellulase secretion can be induced by cellulose intake. Different from the synchronously upregulated cellulase and hemicellulase genes, the laccase gene was significantly downregulated. This gene encoded by the salivary glands of lower termites, was confirmed to function in lignin oxidation and depolymerization [22,39]. The whole body (*p* = 0.02) and endogenous (*p* = 0.03) hydrolytic activity of *C. formosanus* on pyrocatechol (diphenol oxidases substrate) were both significantly inhibited after kojic acid treatment. Laccase belongs to diphenol oxidases [39]. A previous study showed that the laccase activity could be inhibited to 10% of control activity by 20 mM of kojic acid [44]. Thus, the kojic acid treatment should also inhibit laccase activity from the *C. formosanus* salivary gland to a certain extent. Furthermore, inhibition of laccase activity did not lead to compensatory upregulation of laccase gene expression, which was consistent with the substrate (lignin) inducibility of lignin-degrading enzyme secretion in the termites. The results above suggest there to be a close correlation between expressions of cellulase and hemicellulose in *C. formosanus*, while expression regulation of laccase gene was relatively independent.

### 4.2. Possibility of Rare Bacterial Communities Participating in Oxidative Detoxification

In this study, the total relative abundance of Bacteroides and Spirochetes of hindgut accounted for more than 70% of the total bacterial community. Although the kojic acid treatment group exhibited decreased abundances of those phyla, the differences were not considered to be significant. Almost all Bacteroides in *P. grassii* occur within the cell, while a small proportion is attached to the surface. Bacteria (Bacteroides as major) living within *Holomastigotoides* spp. and *S. leidyi* are very rare, accounting for approximately 0.5% of the total bacterial community [12,45]. Thus, after kojic acid treatment, the number of *P. grassii* decreased slightly (*p* > 0.05) with the decrease of Bacteroides and Spirochetes amount (*p* > 0.05). Reversely, the number of *Holomastigotoides* spp. (*p* > 0.05) and *S. leidyi* (*p* = 0.02) were both increased by kojic acid treatment. The single-cell transcriptomes showed overlaps in expression of β-glucosidase, endo-β-1,4-glucanase and xylanase among four flagellates in *C. formosanus*, although these enzymes had differential expression patterns in different protists [7]. Thus, the converse variations in the number of several flagellates further showed their collaboration in the process of adapting to toxic stress. The flagellates play a major role in the glycosylation of lignocellulose in termites (reviewed in [13]). The phyla Bacteroides and Spirochetes are the most important members involved in carbohydrate fermentation and acetic acid reduction reaction in *C. formosanus* [6,46,47]. Therefore, the number of dominant lignocellulose decomposers and energy producers (protist, Bacteroides and Spirochetes) in the hindgut of *C. formosanus* maintained an overall stable abundance regarding these organisms, showing a possible regulation mechanism that allows the tolerance of the termite to low toxic treatments, as termites maintained normal feeding activity and survival rates.

On the other hand, the relative abundances of 10, 9, 27 and 12 taxa from the major phyla Bacteroides, Firmicutes, Proteobacteria, and Actinomyces were significantly increased (*p* < 0.05), respectively. From genus to class level, almost all of these enriched taxa belonged to low abundance microbiota (much less than 1% of total sequencing tags). In addition, the abundance of two rare phyla, *Acidobacteria* and *Chloroflexi*, was increased 425-fold (*p* = 0.024) and 760-fold (*p* = 0.008), respectively, in the kojic acid treatment compared to the control. In general, the diversity of termite bacterial communities is determined by the population of rare microbiota [48]. The number of bacterial species in *C. formosanus* was significantly enhanced by the kojic acid treatment (*p* = 0.005). Environmentally adaptive bacteria usually belong to low abundance lineages, and appear or disappear with environmental changes or special functional requirements [49]. In the natural feeding environment, termites need to metabolize large amounts of lignin in wood. Lignin and its intermediate products are physiotoxic to termites [50]. A proportion of the lignin and small molecular phenol-substances produced during lignin degradation still undergo modification and oxidation in the hindgut. This suggests that hindgut microorganisms are involved in the detoxification [16,42]. The transcriptome analysis also showed that various detoxification genes, including laccases (phenoloxidase activity) gene, were encoded by hindgut microorganisms in *C. formosanus* [8]. The phenoloxidase activity of the hindgut here was slightly affected (*p* > 0.05) by kojic acid treatment. Thus, it can be speculated that these low abundance bacteria, which increased their abundance after exposure to the kojic acid treatment, may undertake the compensation functions required for the inhibition of phenoloxidase activity.

Many of the different bacteria affiliated with the phyla listed above have practical detoxifying oxidative activities. For example, Actinomyces strains with superoxidase laccase/phenol oxidase activity have been isolated [18,51], and bacteria belonging to order Bacillus (Firmicutes) and order Enterobacter (Proteobacteria) with lignin superoxidase activity, have also been found [15], as well as *Burkholderia* spp. and *Citrobacter* spp. affiliated to Proteobacteria isolated from the gut of *C. formosanus* exhibiting the ability to degrade aromatic model compounds [51]. In particular, the order Clostridial (Firmicutes), which showed the largest increase in abundance after kojic acid treatment, is distributed mainly in the high-oxygen and high-alkali regions of the hindgut of termites and are considered to be involved in the oxidative pretreatment of lignin in lignocellulose [52,53]. The interior of flagellate cells and the central area of the termite hindgut are extremely anerobic, and thus lack the oxygen required for the oxidative reaction [10,17]. Therefore, the endo-symbiotic bacteria of flagellates, which are dominated by Bacteroides, should be unable to oxidize/decompose lignin and other toxic substances. The decreases in number of these microorganisms was likely due to their lack of detoxification ability. Based on the above results, we supposed that some rare bacteria affiliated with the phyla Firmicutes, Proteobacteria and Actinomyces in *C. formosanus* might be important participants in the metabolism of toxic substances or some intermediate metabolites of lignin in the hindgut.

## 5. Conclusions

The tolerance responses of lignocellulose dual degradation system of *C. formosanus* of non-lethal toxic stress via kojic acid treatment for 10 days were investigated in this study. Our result showed that the low toxic stress would lead to the synergetic decrease of cellulase and hemicellulase activities and increases of corresponding gene expression. Compared with endogenous cellulase/hemicellulase, symbiotic originated hydrolases in hindgut were more sensitive to the kojic acid treatment. However, this low toxic stress did not change the importance of microbial cellulase as the main force in the lignocellulolytic system of *C. formosanus*. The substrate dependence of hemicellulase and laccase was divergent, indicating an independence of expression regulation between the lignin laccase gene and cellulase/hemicellulase genes in *C. formosanus*. For the hindgut microbial community, the core microbiota responsible for cellulose hydrolysis fermentation and energy conversion of *C. formosannus* (protist and bacterial phyla Bacteroides and Spirochete) were stable regarding its abundance during the tolerance to the toxic treatment. We consider this as an important reason for termites to maintain normal feeding activities and survival rates under toxic stress. Furthermore, a large number of low abundance taxa, mainly affiliated with the phyla Firmicutes and Proteobacteria, had a significantly higher abundance after 10 days of kojic acid treatment. These low abundance bacterial lineages greatly affected the diversity of bacterial community in hindgut and may be involved in the supplement of phenoloxidase activity or oxidative detoxification to allow the adaptation of *C. formosanus* to environmental changes. Above all, these findings contribute new knowledge to our understanding of environmental adaptation of the triple cooperative lignocellulose hydrolysis system (host-protist-bacteria) in termites.

## Figures and Tables

**Figure 1 insects-12-00510-f001:**
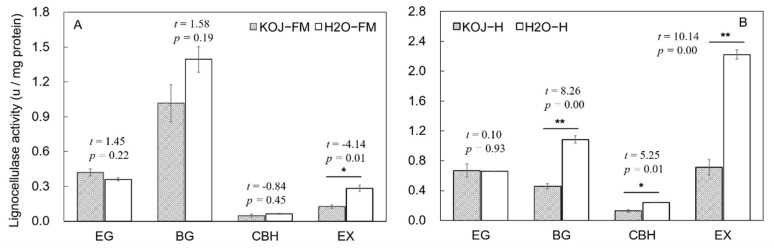
Effect of kojic acid treatment on the lignocellulase activities in the foregut/salivary gland and midgut (**A**) and hindgut (**B**) of *Coptotermes fomosanus*. Data represent the mean ± standard error of the mean (SEM). Asterisks indicate significant differences between the two treatment groups based on the statistical analysis using the *t*-test method (*, *p* < 0.05; **, *p* < 0.01). Abbreviations: FM, foregut/salivary gland and midgut; H, hindgut; KOJ, kojic acid; H_2_O, distilled water; EG, endo-β-1, 4-glucanase, CBH, cellobiohydrolase; BG, β-glucosidase; EX, endo-β-1, 4-xylanase.

**Figure 2 insects-12-00510-f002:**
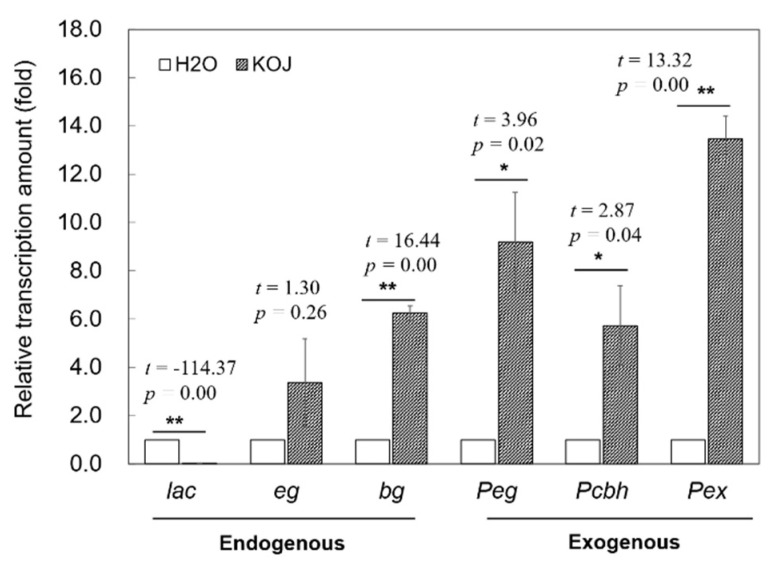
Relative transcription analysis of the six lignocellulase genes in response to kojic acid treatment in *Coptotermes fomosanus*. Termites treated with dH_2_O were used as controls, and the relative expression levels of specific genes were normalized to that of the reference gene *hsp-70*. Asterisks indicate significant differences between the two treatment groups based on statistical analysis using the independent samples *t*-test method (*, *p* < 0.05; **, *p* < 0.01). Abbreviations: lac, *eg* and *bg* symbolize endogenous laccases, endo-β-1, 4-glucanase and β-glucosidase genes, respectively; *Peg*, *Pcbh* and *Pex* symbolize exogenous endo-β-1, 4-glucanase, cellobiohydrolase and endo-β-1, 4-xylanase, respectively; KOJ, kojic acid; H_2_O, distilled water.

**Figure 3 insects-12-00510-f003:**
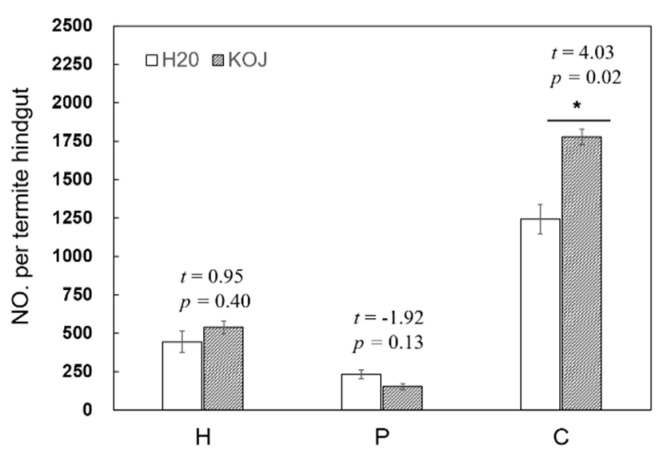
The number of *Pseudotrichonympha grassii*, *Holomastigotoides* spp. and *Cononympha leidyi* per gut of *Coptotermes formosanus* workers after feeding with a control diet (distilled water) or kojic acid (20 mM). Data represent the mean ± standard error of the mean (SEM). Asterisks indicate significant differences between the two treatment groups based on statistical analysis using the independent samples *t*-test method (*, *p* < 0.05). Abbreviations: KOJ, kojic acid; H_2_O, distilled water; H, *Holomastigotoides* spp.; P, *Pseudotrichonympha grassii*; C, *Cononympha leidyi*.

**Figure 4 insects-12-00510-f004:**
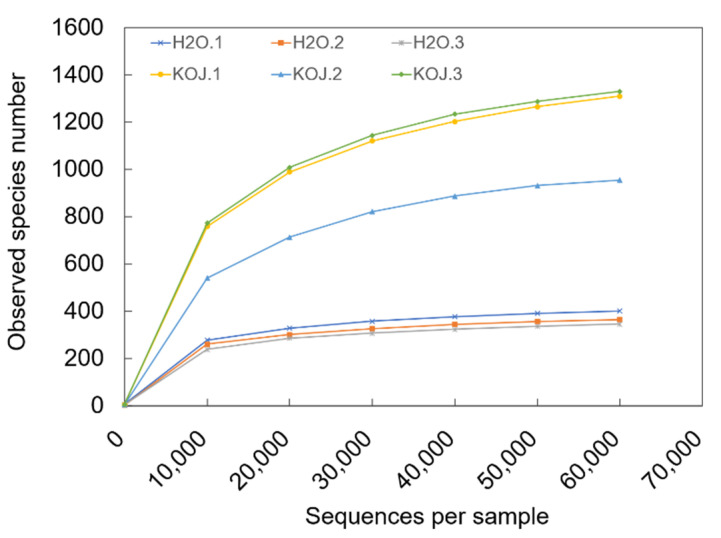
Rarefaction curve of species-abundance data. The data were from 16S rRNA gene library of hindgut bacterial community in *Coptotermes formosanus*. Abbreviations: KOJ, kojic acid; H_2_O, distilled water.

**Figure 5 insects-12-00510-f005:**
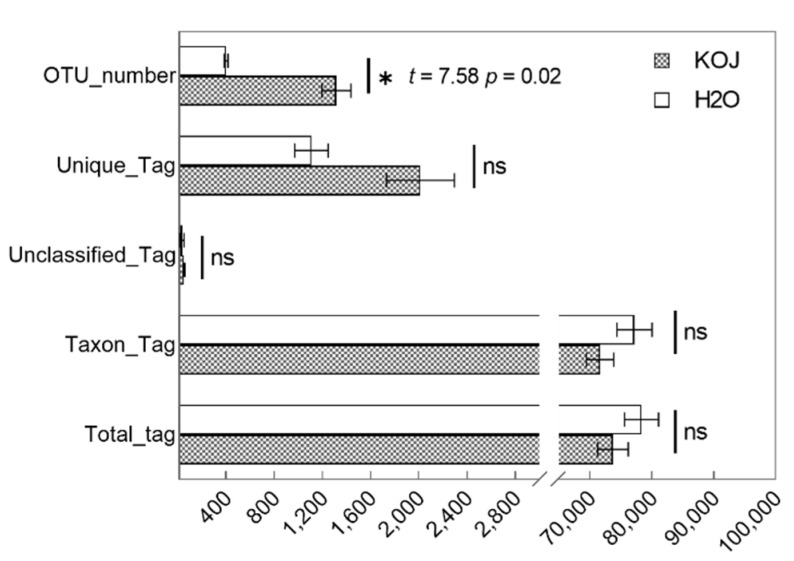
Statistical analysis of absolute sequencing abundance in two groups of *Coptotermes formosanus*. Data represent the mean ± standard error of the mean (SEM). Asterisks indicate significant differences between the two treatment groups based on statistical analysis using the independent samples *t*-test method (*, *p* < 0.05). Abbreviations: KOJ, kojic acid; H_2_O, distilled water.

**Figure 6 insects-12-00510-f006:**
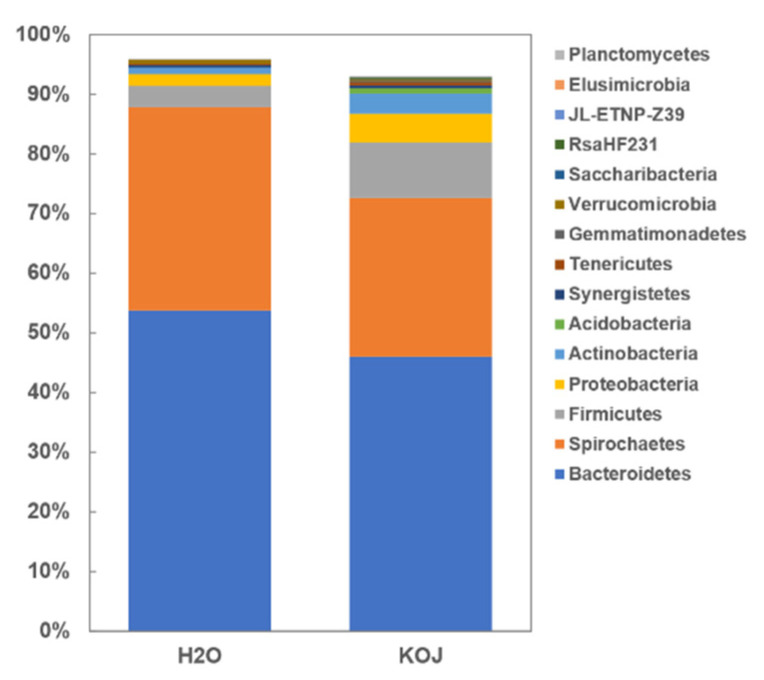
Relative percentage stacked analysis of the 15 most common phyla of the microbial communities in the two termite groups. Abbreviations: KOJ, kojic acid; H_2_O, distilled water.

**Figure 7 insects-12-00510-f007:**
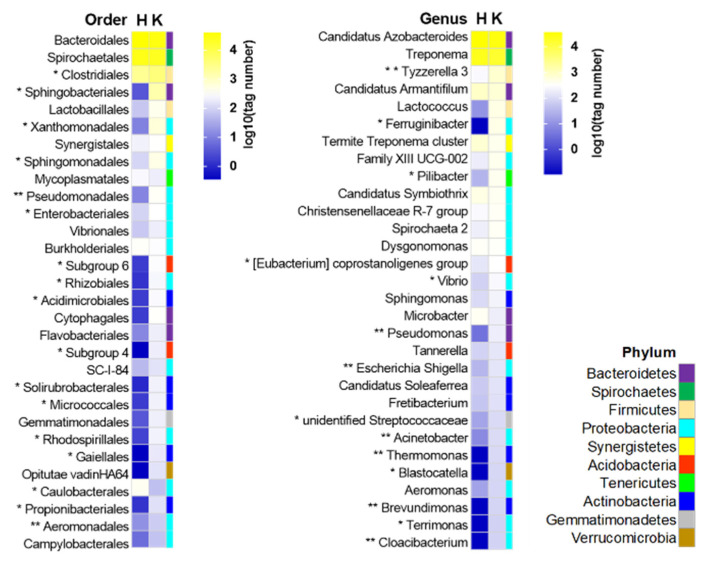
Heat maps of the top 30 most abundant genera and orders in two the termite groups. The absolute abundances of the microbial clusters are presented as varying color intensities. Abbreviations: H, H_2_O control; K, kojic acid treatment; Asterisks indicate significant differences between the two diet treatments (*t*-test and Metastat methods, * and ** for *p* < 0.05 and *p* < 0.01, respectively).

**Table 1 insects-12-00510-t001:** The primers used for quantitative real-time PCR. Abbreviations: *lac*, *eg* and *bg* symbolize endogenous laccases, endo-β-1, 4-glucanase and β-glucosidase genes, respectively; *Peg*, *Pcbh* and *Pex* symbolize exogenous endo-β-1, 4-glucanase, cellobiohydrolase and endo-β-1, 4-xylanase, respectively.

Gene Name	Tissue localization	Primers (5′–3′)	GenBank Accession No.
*lac*	Salivary gland	F: CTCCCGGACATCAACTATCTTC R: GCATAGGATGTCGTCTGGTACA	KX386002.1
*eg*		F: CTGCCATCGCCTACAAGAGTGCT R: GTGTTGTCGTTGGTCGCCCTGTA	EU853671.1
*bg*		F: AGTGGCCCGAGTCTGCTTCTTC R: CGCAGCCATTCGAAGTTGTCTAT	GQ911585.1
*Peg*	Protist: *Pseudotrichonympha grassii*	F: GAGTGAATGTGGTGGAATGGAATC R: GCCATGGAGGATTGACAGGAC	AB071001.1
*Pcbh*		F: TGGACTGAATGGTGCACTCTACTT R: CATTTTCATCATTCGTTTGTGGTT	AB071864.1
*Pex*	Protist: *Holomastigotoides hartmanni*	F: ATCAGTTTTGGAGTGTGCGTCAGG R: TTATATCCGAACTCCCGCTGCTC	AB469374

**Table 2 insects-12-00510-t002:** The inhibitory effect of kojic acid on phenoloxidase of *Coptotermes formosanus*. Data represent the mean ± standard error of the mean (SEM). Significant differences in the alpha diversity between the two groups were based on statistical analysis using the *t*-test method (*p* < 0.05). Abbreviations: WB, whole body of termite, FM, foregut/salivary gland and midgut; H, hindgut.

Assays		Kojic Acid	Distilled Water	*p*-Value
Mortality (%)		4.87 ± 0.47	4.16 ± 0.66	0.75
Feeding amount (mg) ^a^		13.00 ± 0.53	15.34 ± 1.14	0.07
PO activity (U/mg protein)	WB	0.53 ± 0.07	0.91 ± 0.07	0.02
	FS	0.91 ± 0.14	1.38 ± 0.03	0.03
	H	0.28 ± 0.08	0.31 ± 0.04	0.78

^a^: consumption rate is defined as mg of filter of 80 termites per day.

**Table 3 insects-12-00510-t003:** Comparison of the alpha diversity indices between the kojic acid treatment and dH_2_O control Tukey tests (*p* < 0.05) were used to analyze the significant differences in the alpha diversity indices between the two treatment groups. ACE, Abundance Coverage-based Estimator; PD whole tree, PD whole tree’s Index, which reflects the complexity of the species in the community. Data represent the mean ± standard error of the mean (SEM) (*n* = 3).

Alpha Diversity Index	Distilled Water	Kojic Acid	Tukey *p*-Value
Observed species	371 ± 16	1198 ± 121	0.005
ACE	403.82 ± 17.75	1304.89 ± 145.78	0.008
Chao1	403.95 ± 19.64	1290.93 ± 155.52	0.008
PD whole tree	38.99 ± 1.19	91.54.99 ± 6.12	0.004
Shannon	4.15 ± 0.44	5.41 ± 0.25	0.068
Simpson	0.74 ± 0.08	0.84 ± 0.02	0.402

**Table 4 insects-12-00510-t004:** The distribution analysis of the taxa (≥0.1%) with significant changes in relative and absolute abundance. Abbreviations: KOJ, kojic acid; H_2_O, distilled water.

Phylum	Relative Abundance (%)		Absolute Abundance (Tag Number)		Significant Shift Taxa (Number)	
	H_2_O	KOJ	H_2_O	KOJ	Upregulated	Downregulated
Bacteroidetes	53.71	46.01	41899	32913	10	0
Firmicutes *	3.43	9.27	2763	6708	9	0
Proteobacteria *	2.02	4.93	1552	3586	27	0
Actinobacteria *	1.21	3.35	928	2738	12	0
Verrucomicrobia *	0.55	0.25	423	180	0	3
Acidobacteria *	0.00	0.85	2	622	8	0
Chloroflexi *	0.00	0.38	0	273	1	0

*: significant differences between the two treatment groups based on statistical analysis using the *t*-test and Metastat methods (*p* < 0.05). Data represent the mean (*n* = 3).

## Data Availability

The study did not report any data.

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
