# Peer review of "Collaborative Response of the Host and Symbiotic Lignocellulytic System to Non-Lethal Toxic Stress in Coptotermes formosanus Skiraki"

_insects, 2021, doi:10.3390/insects12060510_

Round 1
Reviewer 1 Report
The survival of termites on lignocellulose depends on the synergism of digestive enzymes provided by host termites and their symbiotic microorganisms. Manuscript, entitled " Collaborative response of the host and symbiotic lignocellulytic system to non-lethal toxic stress in Coptotermes formosanus Skiraki" used kojic acid as their key strategy to establish a non-lethal toxic stress model and explore the tolerance mechanism of digestive system in the termite Coptotermes formosanus. In hindgut, responds come mainly from the variety of low abundance bacteria rather than the main microbial community, potentially participate in the detoxification of toxic substances. This study provides new information on the mechanism of termite toxicity tolerance from the perspective of digestive system for termite adaptability. But, some revisions are required for publication of this study.
- P1: Simple Summary need to simplify.
- Line 11: “pacific area” to many areas.
- Line 31: “Mmeanwhile”, spelling mistake. And, “were” to “became”.
- Line 33: After “might” add “be”.
- Lines 50 and 53: “grassii” should be Italic.
- Line 68: “approach”, spelling mistake.
- Line 269: after “spp”, lost dot.
- Line 270: “grassi” is “grassii”.
- Line 305: MRRP changes to MRPP.
- Line 439: pritist is Protist.
- Line 440: abundance is spelling mistake.
- Line 473: after “sp”, lost dot.
- Lines 504 and 515: pritistit is Protist.
- Figures 6, screen-shots come from a bioinformatics platform. It should be replaced by an appropriate figure.
- To Check all science names of bacterial phyla in the
- The supplementary figures are lack of Figure legends.
Reviewer 2 Report
This is work adds new knowledge to the field and has a lot of data and work therein. Despite this, there are some improvements that may be done.
One concern or addition to the discussion: the effect on filter paper on lower termites symbiotic community is evident (ex. Raychoudhury et al. 2013), in the future a control with termites fed on wood would be an interesting addition to this type of studies.
Methods need to be reviewed as some information is missing.
Please add spacings between all numbers and their units throughout the text (for example, for % and temperature this was not done).
I think there is information that would improve the methods understanding by the readers, which is to add that the anterior part of the termite digestive system concentrates host (termite) cellulases, and the hindgut concentrates all symbiotic fauna and their enzymes.
References style and formatting should be thoroughly reviewed according to the journal rules, references are not uniform regarding format.
In the introduction, some future applications are referred, regarding termite control strategies, but in the discussion/conclusions, this is not referred to. I find this a missing link, and the advice would be to remove this from the article, as it does not make any difference for your findings, or add it in the Discussion/conclusions.
Some suggestions and corrections are as follows:
L10 - representative is not necessary here
L11 - add "the" Pacific area.
L13 - correct "survival"
L16 - why using microflora instead of symbiotic community, for example? Are you considering only bacterial cellulases or also protist cellulases?
L19 - analysed taxa and cellulases?? or only cellulases? in this case can you surely trace back to taxa?
L24 - suggest adding "conditions" after "environment"
L31 - erase one "m" in Meanwhile
L32 - add "be" between might and responsible
L49 - "consists of" does not seem to fit here, maybe "harbours"
L50 - put "grassii" in italics and eliminate the italics in "and"; descriptors name should be added to the species name
L53 - put "grassii" in italics
L54 - when first referred to in the text, the species name should have the genera not abbreviated and the descriptors name should be added
L55 - "attached"
L56/57 - please rephrase as it is confusing - the major functions...are the performance of multiple..." for example.
L62 - looks like the number 9 should be included in the parentheses
L63 - add spacing before [10,21]
L68 - approach misspelled
L68-70 - please rephrase this, at least put termiticide in the plural, and "the few existing studies..."
L70 - lignocellulolytic instead of lignocellulose
L82 - replace "to" by "the"
L89 - please refer to the conditions of termite laboratory rearing (temperature and relative humidity) and the origin of the C. formosanus colonies used. Did you use one or more colonies? Please clarify these details.
L92 - please clarify the origin of kojic acid solution (if commercially obtained refer to the company/country)
L96 - that water supplement was added to the filter paper? Inside the petri dishes there were only termites and filter paper, or any other material (sand, for example) was added?
L98 - it is not clear how many replicates per treatment were done, please clarify this in the methodology, it was five or ten? define whether you call group or treatment to the different trials (kojic or water/control)
L99 - instead of feeding amount I suggest feeding rate or consumption rate
L133 - if possible add here a reference
L136 - rephrase please - for example, "for better correlating the inhibition effect with the lignocellulolytic variations, " and replace the final mark with a comma, linking with the other sentence.
L152 - please clarify whether the termites referred here are the same defined above (fed with filter paper) or if they were fed with another source of cellulose, or starved.
L156 - if the primers were designed by you or have a reference paper, please add them in the methodology section, if not please make it clear that they were used according to other authors
L166 - did you used any kind of haemacytometer to do this? If so add this information to the methodology
L167 - at the beginning of the sentences usually the species name should not be abbreviated
L168 - rephrase: "are difficult to distinguish by morphological characters..."
L183 - add bp after 0.1
L198 - this part is more adequate in the statistical methods
L226 - instead of feeding amount I suggest feeding rate or consumption rate; usually this is calculated per termite, do you have any special reason for doing this calculation for 80 termites?
L233 - here you name it control group (which is correct), and it would be useful to add this definition to the methodology section
L243 - abbreviations should ideally be present in all tables and figures captions
L248 - "very" not needed here
L250 - substitute "that in in" by "the"
L267 - add a full stop after spp
L268 - misses an "i" in grassii
L270 - erase full mark after Pseudotrichonympha; spp not in italics and with a full mark in the end
L271-272 - this sentence may be erased, as it is already described above
L279 - rank abundance or rarefaction curves?
L283 - please complete the caption of the figure with the type of organism analyzed, termite, etc
L285 - table 3 should appear before Figure 5; add full stop after Fig
L286 - replace OUT by OTU
L287 - discard the word "very"
L320 - "67 and three bacterial taxa" is not clear its meaning, if possible please rephrase this referring to "more or less"
L336 - "compared with..." instead of "compared with that in the..."
L343 - "Generally, a 20 mM kojic acid treatment for 10 days did not reshape significantly the bacterial community...
L352 - is it possible to add SE to the mean values? Also, put the 2 from H2O inferior to the line
L353 - not sure the meaning of this: "In this study, the filter paper fed termites belongs to pure cellulose", please rephrase it, as it is confusing
L382 - discard the parentheses at the end of the sentence
L394 - rephrase for example like: "the activity of exogenous EX (p = 0.001) significantly decreased the most after the kojic acid treatment"
L411 - could you explain better the connection between laccase in white rot and the termite salivary glands? Is it linked with lignin degradation?
L412-414 - please rephrase "This is...secretion" it is confusing
L419 - add an "s" to Bacteroide
L420-421 - the phrase is also confusing, rephrase it, for example: "Although the kojic acid treatment group exhibited decreased abundances of those groups, the differences were not considered to be significant"
L422 - add "a" before small; and "is attached" instead of "attach"
L423 - bacteria are not parasitized by protists, this must be clear. Do you mean bacteria living inside Holomastigotoides? And if you want to refer to Bacteroides, please add that information here too, to clarify the sentence
L425 - so the hypothesis is that the decrease of bacteria causes a decrease/increase in different protist species, is that correct? They are correlated, right? I think there is a verb missing before Bacteroides
L430 - not sure about "converse" meaning here, maybe replace by reverse? opposite?
L433 - add an "s" to Bacteroide
L436 - protist is misspelled
L437-439 - please rephrase. suggestion "...maintained an overall stable abundance regarding these organisms, showing a possible regulation mechanism that allows the tolerance of the termite to low toxic treatments, as termites maintained normal feeding activity and survival rates"
L438-439 - this phrase may reinforce the previous, as suggested above
L440 - suggestion: On the other side, the relative abundances of some taxa belonging to the major phyla: Bacteroides (10 out of [add here the total number of taxa of Bacteroides]), Firmicutes (9 out of [total number of taxa of Firmicutes]), Proteobacteria (27 out of [add here the total number of Proteobacteria]), and Actinomyces (12 out of [add here the total number of Actinomyces]) were significantly increased. "
L442 - this sentence is not clear, please rephrase it
L446 - replace "with that in" by "to"
L447/8 - would rather be better to use microbiota here, instead of microflora; discard "We found" and "very"; replaced "enriched" with "enhanced"
L452 - how do you explain that a rare species is a keystone taxon? this statement has to be developed in order to make sense. Does it perform any key function?
L453 - the reference is not a number, please correct; Erase the "And"; maybe link the previous and this phrase, and the answer to the previous question is here.
L463/4 - good, just adjust the text as suggested: "...that these low abundance bacteria, which increased their abundance after exposure to the kojic acid treatment may undertake..."
L469 - orders name is not in italics
L470 - add a full mark after sp.
L479 - replace "That is to say" by "Probably"
L484 - please rephrase, as it is not clear the message
L485/8 - Also needs rephrasing to be clearer
L490 - replace "to" by "of"
L496 - changed instead of change
L497 - erase "And"
L500 - better use microbiota instead of microflora
L501 - protist is misspelled; replace the corresponding part of the text by: "...were stable regarding its abundance..."
L505 - replace enriched with another expression (increased, for example), the correct would be "had a significantly higher abundance after..."
L506 - which treatment? I know, but this should be clearly stated in the text to prevent misunderstandings
L508 - ..."to allow the adaptation of C. formosanus to..."
L509 - in a small text you refer to the bacteria as rare, non-major and low abundance taxa, it would be advantageous to stick to one or two adjectives instead of 3.
L510 - "... contribute to add new knowledge..."
L512 - protist misspelled
Reviewer 3 Report
The manuscript by Zeng et al is a nice contribution describing influence of non-lethal toxic on termite physiology and its symbiotic systems. The authors major conclusion is well supported, however. there are some questions and points to improve. They've only seen the impact of kojic acid for 10 days, is that enough to discuss? The long-term impact should also be mentioned. I couldn't understand why there is no change in the relative numbers of taxa and absolute aboundance (Table 4, Fig6) despite the predominant increase in the number of OTUs (Fig. 5)? Minor points The color of the columns in all figures should be the same for the control and KOJ treatment areas. I don't understand the meaning of the symbol (M, P, S) in Figure 3, so please correct it. For Figure 2, not only average values but the distributions (or standard errors or values) of Control is also important, so both control (H2O) data and KOJ treatment data should be shown in parallel like other graphs.Author Response
Please see the attachment
